# Pineapple Peel Flours: Drying Kinetics, Thermodynamic Properties, and Physicochemical Characterization

Carolaine Gomes dos Reis [1], Rossana Maria Feitosa de Figueirêdo [1], Alexandre José de Melo Queiroz [1],
Yaroslávia Ferreira Paiva [1,*], Lumara Tatiely Santos Amadeu [1], Francislaine Suelia dos Santos [1],
João Paulo de Lima Ferreira [1], Thalis Leandro Bezerra de Lima [1], Fabrícia Santos Andrade [1],
Josivanda Palmeira Gomes [1], Wilton Pereira da Silva [1] and Dyego da Costa Santos [2]

1 Department of Agricultural Engineering, Federal University of Campina Grande,
  Campina Grande 58429-900, Brazil; carolainetecalimentos@gmail.com (C.G.d.R.);
  rossana.maria@professor.ufcg.edu.br (R.M.F.d.F.); alexandrejmq@gmail.com (A.J.d.M.Q.);
  lumaratatielyea@gmail.com (L.T.S.A.); francislainesuelis@gmail.com (F.S.d.S.);
  joaop_l@hotmail.com (J.P.d.L.F.); tthallisma@gmail.com (T.L.B.d.L.);
  fabricia.santos.andrade@hotmail.com (F.S.A.); josivanda@gmail.com (J.P.G.); wiltonps@uol.com.br (W.P.d.S.)
2 Federal Institute of Education, Science and Technology of Rio Grande do Norte,
  Paus dos Ferros 59900-000, Brazil; dyego.csantos@gmail.com
* Correspondence: yaroslaviapaiva@gmail.com

**Abstract:** Pineapple is a widely cultivated, consumed, and processed fruit by the industry. However, only 22.5% of the whole fruit is used, which constitutes economic waste and environmental impact. The objective was to determine the drying kinetics and characterize the residual peel flours of two pineapple varieties at four drying temperatures. *Jupi* and *Pérola* pineapple peels were dried at temperatures of 50, 60, 70, and 80 °C in a thin layer. Ten mathematical models were adjusted to the experimental data to characterize the drying process. Fresh samples and flours were characterized according to their physicochemical properties (water content, ash, water activity, total sugars, reducers, pH, acidity, proteins, lipids, carbohydrates, and total energy value—TEV). The Midilli model was chosen because it best represents the drying process with high values of determination coefficients ($R^2$) and low mean squared deviations (MSD), Chi-square ($\chi^2$), and estimated mean error (EME). The increase in temperature led to an increase in the effective diffusivity coefficient and consequent reduction in drying time. The activation energy obtained from the Arrhenius equation was 24.59 and 26.25 kJ/mol for *Jupi* and *Pérola*, respectively. Differences were reported in the enthalpy and entropy decrease with the increasing temperature, contrary to the Gibbs free energy. The flours produced had good characteristics for conservation, being acidic with low water content and low water activity. High levels of total and reducing sugars, carbohydrates, and total energy value were observed, in addition to good protein content.

**Keywords:** *Ananas comosus*; agricultural residues; sustainability; dehydration

## 1. Introduction

Pineapple (*Ananas comosus* (L.) Merryl) is appreciated all over the world, possibly as the most economically important bromeliad fruit. Brazil is one of the leading countries in pineapple production, with a production of 1617.684 tons in 2019, with the main producers being the states of Pará (311,947 tons), Paraíba (307,116 tons), and Minas Gerais (179,287 tons) [1]. Pineapple farming in Paraíba has always been relevant in national production, with pineapple being one of the most attractive tropical fruits. In fruit growing in Paraíba, it is the crop that has the greatest economic significance, with a high demand for labor caused by the lack of mechanization of the crop [2].

Pineapple is usually consumed in its natural form or in the production of juices. However, other products are made in smaller proportions, such as pineapple syrup, jelly,

jam, and dehydrated pineapple [3]. Pineapple waste flours are developed and tested in innovative food products, such as vinegar, vanillin [4], and bread [5]. Research shows that drying has effects on the physical–chemical and technofunctional properties in the preparation of pineapple peel flour, showing functional properties that enable its use in the form of flour rich in dietary fiber, thus contributing to the environment [6].

Fruit processing generates a significant amount of waste, which can be used in the production of new products, as they usually have significant nutritional value [7]. The pineapple canning industry is one of the many food industries that contribute to the accumulation of solid waste [8]. About three quarters of the pineapple, comprising the peel, stalk, and crown, are classified as waste by the fruit pulp processing industries [9]. This high amount of organic material receives inadequate disposal, thus generating economic and nutritional waste [10], since the composition of the different types of waste such as the bark, core, crown, stem, and residual pulp have a nutritional composition such as carbohydrates (50–80%), vitamin C, and beta-carotene, which makes it suitable for the development of value-added products [11,12].

Drying is a conservation and transformation technology that can be adopted to take advantage of pineapple residues, transforming them into a new product. Drying is conventionally defined as a process of removing water from agricultural products that involves the simultaneous transfer of heat and mass [13], providing conservation of the material during storage [14]. Through drying kinetics, the data obtained allow us to understand and describe the behavior of the product and its interactions with mass transfer and the drying agent, whose particularities associated with the diversity of materials of biological origin stimulate the interest of researchers in analyzing the drying behavior of the most varied products [15].

Lately, pineapple waste is dried in different ways: pineapple peels enriched by fermentation and dried via forced air circulation at different temperatures and air speeds [16]; dried via refraction at different temperatures [6], and dried in vacuum [17]. The best drying method depends on the purpose attributed to the product, since freeze-drying was more effective in preserving anthocyanins, and oven drying showed higher individual phenolic acids, as observed in dried "BRS Magna" grape skins [18]. Convective drying with forced circulation with hot air is the technique most used by chemical industries to dry foods [19]. This method has the advantage of not being pollutive, in addition to being highly efficient in the drying process [20]. The advantages and relative low cost associated with this type of drying juxtapose the transformations of the food, which can be minimized by associating techniques for different materials in particular [21]. It is an artificial method that makes it possible to control the temperature, air flow, and exposure time [22], depending on the final objective. Low temperatures can be used, which allows for the drying of thermosensitive products or products that undergo oxidation quickly [23]. Fruit waste, for example, must be dried under controlled temperature conditions to avoid nutrient degradation [24].

Among the phenomena studied in drying, effective diffusivity is an indication of the flow of water removed from the product during the process [25]. From the determination of the effective diffusivity, the main thermodynamic properties of the dehydration process are calculated, such as the enthalpy, entropy, and Gibbs free energy [26]. According to Corrêa et al. [27], knowledge of thermodynamic properties in the drying process is of fundamental importance as these sources of information are necessary to understand the processes involved such as equipment design, required energy, the properties of adsorbed water, the evaluation of the microstructure of products, and the study of physical phenomena that occur on their surfaces.

Given the above and considering the importance of studies that contribute to the reduction in waste from fruit processing, as well as the comparison of the physico-chemical characteristics of different varieties of pineapple in relation to the drying process, this study aimed to prepare flours from the peel of pineapples of the *Pérola* and *Jupi* varieties from convective drying at temperatures of 50, 60, 70, and 80 °C, to determine the drying kinetics,

to adjust the mathematical models to the experimental data, to determine the effective diffusivity and thermodynamic properties, and to characterize the flours obtained.

## 2. Materials and Methods

### 2.1. Samples Preparation

To prepare the samples, ripe pineapples of the *Pérola* and *Jupi* varieties were cultivated in the municipalities of Itapororoca (Geographical coordinates: 6°49′48″ S, 35°14′49″ W) and Pedras de Fogo (Geographical coordinates: 7°24′7″ S, 35°6′57″ W), respectively, in the state of Paraiba-Brazil. Pineapples were considered ripe if they had °Brix levels equal to or greater than 12° [28], specific weight around 1.012 g/mL [29], greater translucency, and low acidity levels, being correlated with the skin tones, presenting ripeness indices larger in colorful coloring [30]. The pineapples were washed and sanitized in a chlorine solution (200 ppm) for 15 min. The peels were removed using stainless steel knives and ground in a pulper (Laboremus®, Campina Grande, Brazil). The crushed peels were packed in low-density polyethylene bags and stored in a freezer at −18 °C until use, remaining in the freezer for 2 months before starting to dry.

### 2.2. Drying

The pineapple residues (crushed peels) were thawed under refrigeration at 4 °C and after thawing, the samples were left at room temperature for 30 min. The residues were arranged in a thin layer (0.5 cm) on stainless steel trays and dried in an oven with forced air circulation-convective drying (FANEM LTDA, Sao Paulo, Brazil, model 320/5 200 °C) at temperatures of 50, 60, 70, and 80 °C and with a drying air speed of 1.0 m/s.

The drying kinetics were performed by weighing the trays with the samples at regular intervals of 5, 10, 20, 30, and 60 min until reaching the equilibrium water content, then the dry mass was determined according to the Adolfo Lutz Institute [31] and the water content ratio was calculated according to Equation (1).

$$\mathrm{MR} = \frac{X - X_e}{X_i - X_e} \tag{1}$$

where:

　　MR—ratio of the water content of the product (dimensionless);
　　$X$—water content at time t (% db);
　　$X_i$—initial water content of the product (% db);
　　$X_e$—product balance water content (% db).

The mathematical models in Table 1 were adjusted to the experimental data of the drying kinetics using the computational program Statistica version 7.0 via non-linear regression and the Quasi-Newton method.

**Table 1.** Mathematical models used to estimate the drying kinetics of pineapple skin.

| Model Name | Equation | Reference |
|---|---|---|
| Diffusion Approach | $\mathrm{MR} = a \exp(-kt) + (1-a) \exp(-kbt)$ | Sharaf-Elden et al. [32] |
| Two Terms | $\mathrm{MR} = a \exp(-k_0 t) + b \exp(-k_1 t)$ | Henderson [33] |
| Two-term exponential | $\mathrm{MR} = a \exp(-kt) + (1-a) \exp(-kat)$ | Sharaf-Elden et al. [32] |
| Midilli | $\mathrm{MR} = a \exp(-kt^n) + bt$ | Midilli et al. [34] |
| Page | $M = R\exp(-kt^n)$ | Page [35] |
| Henderson and Pabis | $\mathrm{MR} = a \exp(-kt)$ | Henderson & Pabis [36] |
| Logarithmic | $\mathrm{MR} = a \exp\exp(-kt) + c$ | Yagcioglu et al. [37] |
| Newton | $\mathrm{MR} = \exp\exp(-kt)$ | Lewis [38] |
| Verma | $\mathrm{MR} = a \exp(-kt) + (1-a) \exp(-k_1 t)$ | Verma et al. [39] |
| Wang and Singh | $\mathrm{MR} = 1 + (at) + (bt^2)$ | Wang & Singh [40] |

MR—ratio of water content of the product (dimensionless); $t$—drying time (min); $a$, $b$, $c$, $k$, $k_0$, $k_1$, $n$—model constants.

As a criterion to evaluate the quality of the models' adjustments to the experimental data, the coefficient of determination ($R^2$), mean squared deviation (MSD) (Equation (2)), Chi-square ($\chi^2$) (Equation (3)), and the estimated mean error (EME) (Equation (4)) were used, with the best model being considered the one that presents a high $R^2$ value and low values of MSD, $\chi^2$ e EME.

$$R^2 = \frac{\sum_{i=1}^{N} \left[ (MR_{exp,i} - \overline{MR_{exp,i}})(MR_{pred,i} - \overline{MR_{pred,i}}) \right]^2}{\sum_{i=1}^{N} \left( \sum_{i=1}^{N} \left[ \left( MR_{exp,i} - \overline{MR_{exp,i}} \right)^2 \right) \left( \sum_{i=1}^{N} \left( (MR_{pred,i} - \overline{MR_{pred,i}})^2 \right) \right.}$$

(2)

where:

MSD—mean squared deviation;
$RX_{pred}$—ratio of water content predicted by the model;
$RX_{exp}$—experimental water content ratio;
n—number of observations.

$$\chi^2 = \frac{1}{GLR} \sum_{i=1}^{N} \left( MR_{pred,i} - MR_{exp,i} \right)^2$$

(3)

where:

$\chi^2$—Chi-square;
$RX_{pred}$—ratio of water content predicted by the model;
$RX_{exp}$—experimental water content ratio;
N—number of experimental observations;
n—number of model constants.

$$DQM = \frac{\sqrt{\Sigma \left( RX_{pred} - RX_{exp} \right)^2}}{n}$$

(4)

where:

EME—estimated mean error.

### 2.3. Effective Diffusivity

To determine the effective diffusivity, the mathematical model of liquid diffusion was adjusted to the experimental data, considering the uniform initial water distribution and absence of thermal resistance. Equation (5) is the analytical solution to Fick's second law, applied to products with approximate shape to an infinite flat plate. The Arrhenius-type equation (Equation (6) was applied to evaluate the influence of temperature on the effective diffusivity.

$$MR = \frac{8}{\pi^2} \sum_{n=0}^{\infty} \frac{1}{(2n+1)^2} \exp \left[ -(2n+1)^2 \pi^2 D_{ef} \frac{t}{L^2} \right]$$

(5)

where:

MR—ratio of the water content of the product (dimensionless);
$D_{ef}$—effective diffusivity ($m^2/s$);
n—number of terms;
L—characteristic dimension (half plate thickness);
t—time (s).

$$D_{ef} = D_0 \exp \left( \frac{E_a}{RT_a} \right)$$

(6)

where:

$D_{ef}$—effective diffusivity ($m^2/s$);
$D_o$—pre-exponential factor ($m^2/s$);
$E_a$—activation energy (J/mol);
R—universal gas constant (8.314 J/mol K);
Ta—absolute temperature (K).

### 2.4. Thermodynamic Properties

The thermodynamic properties of enthalpy, entropy, and Gibbs free energy, related to the drying process of pineapple residues, were determined according to Equations (7)–(9), respectively [41].

$$\Delta H = E_a - RT \tag{7}$$

$$\Delta S = R \left[ \ln(D_0) - \ln\left(\frac{K_B}{h_P}\right) - \ln(T) \right] \tag{8}$$

$$\Delta G = \Delta H - (T)\Delta S \tag{9}$$

where:

$\Delta H$—enthalpy (J/mol);

$\Delta S$—entropy (J/mol K);

$\Delta G$—Gibbs free energy (J/mol);

$E_a$—activation energy (J/mol);

R—universal gas constant (8.314 J/mol K);

$K_B$—Boltzmann constant ($1.38 \times 10^{-23}$ J/K);

$h_p$—Planck constant ($6.626 \times 10^{-34}$ J/s);

T—absolute temperature (K).

### 2.5. Physicochemical Evaluation and Centesimal Composition of in Natura Peels and Flours

After drying, the dehydrated residues were crushed in a knife mill to transform them into flour.

Samples of fresh pineapple skins and elaborated flours were characterized by means of physicochemical analyses and centesimal composition, in triplicate, with respect to the following parameters: the gravimetric method was used to determine the water content, which consists of the difference in the initial and final weight of the sample after weighing until a constant weight in a vacuum oven at 70 °C, with the results expressed as a percentage (%); the ash samples were incinerated at 550 °C and the results were measured in percentage; for total titratable acidity, the titrimetric method was used with a 0.1 mol/L NaOH solution and phenolphthalein as a turning point indicator and the result was expressed as a percentage of citric acid (%) and proteins, according to the methodologies of Instituto Adolfo Lutz [31]; the water activity ($a_w$) was determined at 25 °C (Aqualab 3TE); the dinitrosalicylic acid (DNS) method was used to determine the levels of reducing sugars in the samples [42]; total sugars were determined using an anthrone solution [43]; lipids were quantified using the method that uses a cold extraction of lipids [44] carbohydrates via difference (100% − (water content + proteins + lipids + ash)) [45]; and the total energy value (TEV) was calculated using the results of the following analyses multiplied by the conversion values: carbohydrates (4 kcal/g), proteins (4 kcal/g), and lipids (9 kcal/g), which were then added up [46].

The data from the physicochemical analyses and centesimal composition were submitted to an analysis of variance (ANOVA) in a factorial scheme of 5 treatments (*in natura*, 50, 60, 70, and 80 °C) × 2 varieties (*Pérola* and *Jupi*) with three repetitions, with the averages compared applying Tukey's test at a 5% probability level using the program Assistat, version 7.7 [47].

## 3. Results and Discussion

Tables 2 and 3 show the mean squared deviations (MSD), coefficients of determination ($R^2$), Chi-square ($\chi^2$), and estimated mean errors (EME) of the models adjusted to the experimental data on the drying of the peel of pineapples *Jupi* and *Pérola*, respectively.

**Table 2.** Values of mean square deviations (MSD), coefficients of determination ($R^2$), Chi-square ($\chi^2$), and estimated mean errors (EME) of the models adjusted to experimental data on drying the peel of pineapple cv. *Jupi*.

| Models | T (°C) | $R^2$ | DQM | $\chi^2$ | EME |
|---|---|---|---|---|---|
| Diffusion Approach | 50 | 0.9992 | 0.0142 | 0.0002 | 0.0148 |
| | 60 | 0.9986 | 0.0186 | 0.0004 | 0.0196 |
| | 70 | 0.9993 | 0.0128 | 0.0002 | 0.0135 |
| | 80 | 0.9989 | 0.0158 | 0.0003 | 0.0168 |
| Two Terms | 50 | 0.9956 | 0.0327 | 0.0012 | 0.0347 |
| | 60 | 0.9924 | 0.0434 | 0.0021 | 0.0464 |
| | 70 | 0.9940 | 0.0374 | 0.0016 | 0.0403 |
| | 80 | 0.9924 | 0.0417 | 0.0021 | 0.0453 |
| Page | 50 | 0.9992 | 0.0143 | 0.0002 | 0.0147 |
| | 60 | 0.9989 | 0.0166 | 0.0003 | 0.0171 |
| | 70 | 0.9995 | 0.0112 | 0.0001 | 0.0116 |
| | 80 | 0.9993 | 0.0131 | 0.0002 | 0.0136 |
| Newton | 50 | 0.9935 | 0.0398 | 0.0016 | 0.0404 |
| | 60 | 0.9888 | 0.0525 | 0.0028 | 0.0534 |
| | 70 | 0.9907 | 0.0465 | 0.0022 | 0.0473 |
| | 80 | 0.9890 | 0.0503 | 0.0026 | 0.0513 |
| Henderson and Pabis | 50 | 0.9956 | 0.0327 | 0.0011 | 0.0336 |
| | 60 | 0.9924 | 0.0434 | 0.0020 | 0.0448 |
| | 70 | 0.9940 | 0.0374 | 0.0015 | 0.0387 |
| | 80 | 0.9924 | 0.0417 | 0.0019 | 0.0434 |
| Two Terms Exponential | 50 | 0.9932 | 0.0409 | 0.0018 | 0.0421 |
| | 60 | 0.9886 | 0.0530 | 0.0030 | 0.0547 |
| | 70 | 0.9905 | 0.0470 | 0.0024 | 0.0487 |
| | 80 | 0.9886 | 0.0511 | 0.0028 | 0.0532 |
| Logarithmic | 50 | 0.9968 | 0.0279 | 0.0008 | 0.0292 |
| | 60 | 0.9944 | 0.0373 | 0.0015 | 0.0392 |
| | 70 | 0.9957 | 0.0316 | 0.0011 | 0.0333 |
| | 80 | 0.9949 | 0.0343 | 0.0013 | 0.0365 |
| Midilli | 50 | 0.9983 | 0.0203 | 0.0005 | 0.0215 |
| | 60 | 0.9993 | 0.0136 | 0.0002 | 0.0146 |
| | 70 | 0.9996 | 0.0097 | 0.0001 | 0.0105 |
| | 80 | 0.9995 | 0.0110 | 0.0001 | 0.0120 |
| Verma | 50 | 0.9992 | 0.0142 | 0.0002 | 0.0148 |
| | 60 | 0.9985 | 0.0191 | 0.0004 | 0.0200 |
| | 70 | 0.9907 | 0.0465 | 0.0024 | 0.0491 |
| | 80 | 0.9989 | 0.0157 | 0.0003 | 0.0167 |
| Wang and Singh | 50 | 0.9625 | 0.0951 | 0.0099 | 0.0993 |
| | 60 | 0.9795 | 0.0709 | 0.0056 | 0.0745 |
| | 70 | 0.9761 | 0.0744 | 0.0059 | 0.0771 |
| | 80 | 0.9822 | 0.0638 | 0.0044 | 0.0664 |

Drying kinetics are usually very well predicted by the models available in the literature, obtaining adjustments with determination coefficients greater than 0.99 and Chi-squares close to zero. Among the adjusted mathematical models, the one by Wang and Singh had the worst results, highlighting the temperature of 50 °C for the *Pérola* variety, with $R^2$ lower than 0.98 and higher values of MSD, $\chi^2$, and EME, for both pineapple varieties.

**Table 3.** Values of mean square deviations (MSD), coefficients of determination ($R^2$), Chi-square ($\chi^2$), and estimated mean errors (EME) of the models adjusted to experimental data on drying the peel of pineapple cv. *Pérola*.

| Models | T (°C) | $R^2$ | DQM | $\chi^2$ | EME |
|---|---|---|---|---|---|
| Diffusion Approach | 50 | 0.9994 | 0.0121 | 0.0002 | 0.0127 |
| | 60 | 0.9988 | 0.0169 | 0.0003 | 0.0177 |
| | 70 | 0.9987 | 0.0180 | 0.0004 | 0.0190 |
| | 80 | 0.9992 | 0.0136 | 0.0002 | 0.0144 |
| Two Terms | 50 | 0.9969 | 0.0272 | 0.0008 | 0.0288 |
| | 60 | 0.9937 | 0.0394 | 0.0018 | 0.0420 |
| | 70 | 0.9918 | 0.0448 | 0.0023 | 0.0482 |
| | 80 | 0.9928 | 0.0405 | 0.0019 | 0.0441 |
| Page | 50 | 0.9993 | 0.0129 | 0.0002 | 0.0133 |
| | 60 | 0.9990 | 0.0155 | 0.0003 | 0.0160 |
| | 70 | 0.9991 | 0.0148 | 0.0002 | 0.0153 |
| | 80 | 0.9995 | 0.0109 | 0.0001 | 0.0113 |
| Newton | 50 | 0.9954 | 0.0331 | 0.0011 | 0.0336 |
| | 60 | 0.9904 | 0.0484 | 0.0024 | 0.0491 |
| | 70 | 0.9874 | 0.0553 | 0.0032 | 0.0563 |
| | 80 | 0.9891 | 0.0499 | 0.0026 | 0.0508 |
| Henderson and Pabis | 50 | 0.9969 | 0.0272 | 0.0008 | 0.0279 |
| | 60 | 0.9937 | 0.0394 | 0.0016 | 0.0406 |
| | 70 | 0.9918 | 0.0448 | 0.0022 | 0.0464 |
| | 80 | 0.9928 | 0.0405 | 0.0018 | 0.0422 |
| Two Terms Exponential | 50 | 0.9951 | 0.0342 | 0.0012 | 0.0352 |
| | 60 | 0.9899 | 0.0498 | 0.0026 | 0.0514 |
| | 70 | 0.9868 | 0.0565 | 0.0034 | 0.0585 |
| | 80 | 0.9889 | 0.0504 | 0.0028 | 0.0525 |
| Logarithmic | 50 | 0.9978 | 0.0231 | 0.0006 | 0.0241 |
| | 60 | 0.9952 | 0.0342 | 0.0013 | 0.0359 |
| | 70 | 0.9940 | 0.0381 | 0.0016 | 0.0403 |
| | 80 | 0.9950 | 0.0339 | 0.0013 | 0.0360 |
| Midilli | 50 | 0.9995 | 0.0114 | 0.0001 | 0.0120 |
| | 60 | 0.9993 | 0.0135 | 0.0002 | 0.0144 |
| | 70 | 0.9993 | 0.0130 | 0.0002 | 0.0140 |
| | 80 | 0.9996 | 0.0096 | 0.0001 | 0.0104 |
| Verma | 50 | 0.9994 | 0.0122 | 0.0002 | 0.0127 |
| | 60 | 0.9988 | 0.0169 | 0.0003 | 0.0178 |
| | 70 | 0.9987 | 0.0180 | 0.0004 | 0.0191 |
| | 80 | 0.9992 | 0.0137 | 0.0002 | 0.0145 |
| Wang and Singh | 50 | 0.9498 | 0.1085 | 0.0124 | 0.1115 |
| | 60 | 0.9739 | 0.0795 | 0.0067 | 0.0821 |
| | 70 | 0.9789 | 0.0713 | 0.0055 | 0.0739 |
| | 80 | 0.9773 | 0.0718 | 0.0056 | 0.0747 |

It was found that, with the exception of the Wang and Singh model, all the applied models satisfactorily adjusted to the experimental data, with emphasis on Diffusion Approximation, Page, and Midilli, with the highest determination coefficients ($R^2 \geq 0.998$) and the lowest MSD ($\leq 0.0203$), $\chi^2 \leq 0.0005$, and EME $\leq 0.0215$, thus indicating an excellent description of the drying process of pineapple skins under the studied conditions. A similar behavior was verified by Barbosa and Lobato [48], in studies of the drying kinetics of pineapple slices, observing $R^2$ values greater than 0.990 when using the Page model in the drying kinetics at temperatures of 60, 65, and 70 °C; and by Olanipekun et al. [49], in drying pineapple slices in a convective dryer with an air velocity of 1.5 m/s at temperatures of 50, 60, and 70 °C; also noting that the Page and Two Terms models presented satisfactory

adjustments ($R^2 > 0.980$ and $\chi^2 \leq 0.0011$). Santos et al. [50], when studying the drying of grapefruit peels, reported on the eleven adjusted models, with emphasis on the Page, Logarithmic, Diffusion Approximation, and Midilli models for all temperatures (60, 70, 80, and 90 °C). However, due to its simplicity, they adopted Page's model as the best representation.

Figure 1 shows the experimental points with the adjustment curves obtained with the Midilli model.

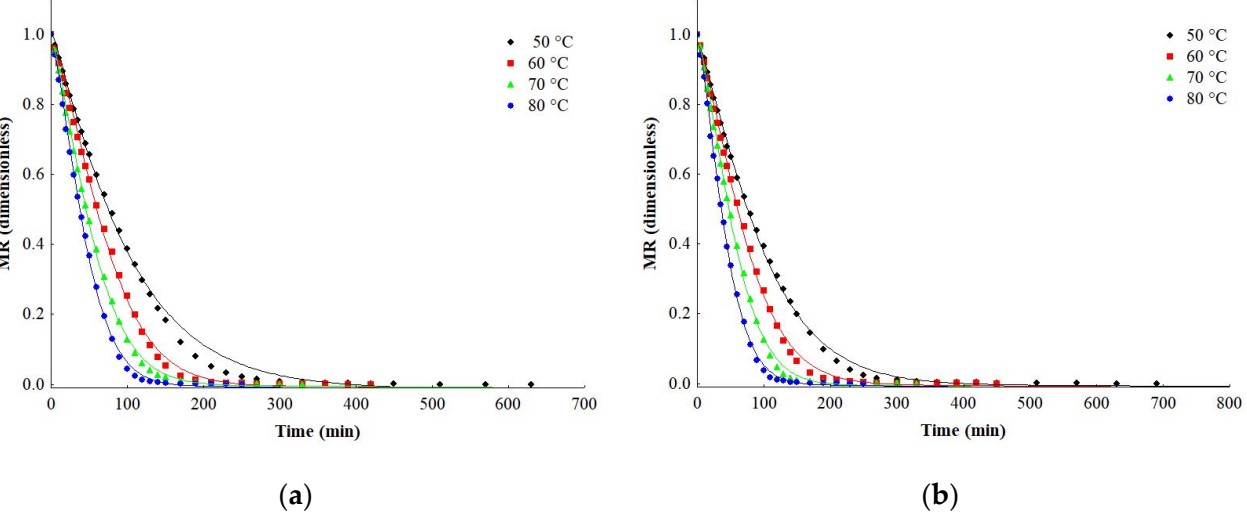

(**a**)　　　　　　　　　　　　　　　　　　　　　(**b**)

**Figure 1.** Experimental and estimated values of the water content ratio (MR) as a function of drying time of pineapple skins with adjustment curves by the Midilli model: (**a**) *Jupi* and (**b**) *Pérola*.

The effect of increasing the drying temperature on the faster reduction in the water content ratio is observed in both varieties with very different curves, demonstrating higher drying rates at higher temperatures. The gradual effect of the increase in temperature on the increase in rates is reported for most varied products, such as the drying of pomegranate by-products at temperatures of 50, 60, 70, and 80 °C with an air velocity of 2 m/s [51]; *jabuticaba* (*Myrciaria jaboticaba*) peels with an air velocity of 5.6 m/s at temperatures of 40, 50, 60, and 70 °C [52]; and dried passion fruit peels at 40, 50, 60, and 70 °C with an air velocity of 1 m/s [53].

The drying time of pineapple peels varied between 630 and 250 min for the *Jupi* variety and between 690 and 250 min for *Pérola* between temperatures of 50 and 80 °C, respectively, with very similar values between the varieties.

The parameters of the Midilli model adjusted to the pineapple peel drying kinetics data are presented in Table 4. The Midilli model was also the one that best adjusted the experimental data of pomegranate peel drying (cv. Wonderful) at drying temperatures of 40, 50, and 60 °C [54].

**Table 4.** Parameters of the Midilli model adjusted to pineapple peel drying kinetics data.

| Variety | Temperature (°C) | a | k | b | n |
|---------|------------------|------|------|------|------|
| *Jupi* | 50 | 1.0136 | 0.0052 | −0.000025 | 1.1380 |
|         | 60 | 0.9695 | 0.0017 | −0.000016 | 1.4498 |
|         | 70 | 0.9819 | 0.0036 | −0.000015 | 1.3729 |
|         | 80 | 0.9768 | 0.0038 | −0.000029 | 1.4245 |
| *Pérola* | 50 | 0.9808 | 0.0030 | −0.000010 | 1.2501 |
|          | 60 | 0.9757 | 0.0022 | −0.000014 | 1.3940 |
|          | 70 | 0.9782 | 0.0024 | −0.000021 | 1.4558 |
|          | 80 | 0.9840 | 0.0043 | −0.000023 | 1.4156 |

Parameter "k" represents the drying rate and parameter "n" reflects the internal resistance to drying. It is observed that the constants "k" and "n" of the Midilli model did not show definite trends between each subsequent temperature. However, between the extreme temperatures, 50 and 80 °C, there is an increase in the value of the parameter "k" in the *Pérola* variety. The same is observed in relation to the constant "n" in both varieties. In the *Pérola* variety, where "n" increases with the increase in temperature in the range of 50 to 70 °C, then decreases at 80 °C, but generally maintains the increasing trend. Moreira et al. [55], studying the drying kinetics of *mandacaru* at temperatures of 40, 50, and 60 °C, obtained, in general, an increase in the constant "n" with the increasing temperature, as observed in this research.

The values of the effective diffusivity as a function of the drying temperature of the pineapple peels are shown in Table 5. An increase in the effective diffusivity is observed with the increase in temperature, explained by the decrease in the viscosity of the water with the increase in temperature, which promotes the ease of its removal [56]. Ojediran et al. [57] described effective diffusivity as the rate at which material water is removed from the center of the geometry to the surface. $D_{ef}$ values close to those of pineapple peels were obtained by Mphahlele, Pathare, and Opara [54] when drying pomegranate peels, obtaining effective diffusivity values of $4.05 \times 10^{-10}$, $5.06 \times 10^{-10}$ and $8, 10 \times 10^{-10}$ m$^2$/s at temperatures of 40, 50, 60 °C, respectively; by Nascimento et al. [53] in passion fruit peel drying with and without ultrasound, observing $D_{ef}$ values that ranged from 0.6 to $2.0 \times 10^{-10}$ m$^2$/s without ultrasound and from 1.2 to $2.6 \times 10^{-10}$ m$^2$/s with ultrasound at drying temperatures of 40, 50, 60, and 70 °C; and by Azeez et al. [58], who reported $D_{ef}$ values of $2.53 \times 10^{-10}$, $3.21 \times 10^{-10}$ and $5.00 \times 10^{-10}$ m$^2$/s for temperatures of 50, 60, and 70 °C, respectively, in the drying of tomato slices.

**Table 5.** Effective diffusivity values as a function of drying air temperature (40, 50, 70, and 80 °C) for *Jupi* and *Pérola* pineapple peels.

| Variety | Temperature (°C) | Effective Diffusivity (m$^2$/s) | R$^2$ |
|---------|------------------|----------------------------------|-------|
| *Jupi* | 50 | $3.24 \times 10^{-10}$ | 0.9699 |
| | 60 | $4.29 \times 10^{-10}$ | 0.9623 |
| | 70 | $5.62 \times 10^{-10}$ | 0.9661 |
| | 80 | $7.01 \times 10^{-10}$ | 0.9642 |
| *Pérola* | 50 | $3.19 \times 10^{-10}$ | 0.9740 |
| | 60 | $4.21 \times 10^{-10}$ | 0.9651 |
| | 70 | $5.55 \times 10^{-10}$ | 0.9607 |
| | 80 | $7.33 \times 10^{-10}$ | 0.9645 |

Figure 2 shows the values of the effective diffusivity (Ln $D_{ef}$) as a function of the inverse of the drying temperature (1/T) adjusted via an Arrhenius-type equation.

Table 6 shows the adjustment parameters of the Arrhenius equation applied to the effective diffusivity data of pineapple peels, observing satisfactory results of the determination coefficients ($R^2 > 0.9990$).

**Table 6.** Arrhenius equation adjustment parameters.

| Variety | D$_0$ (m$^2$/s) | E$_a$ (kJ/mol) | R$^2$ |
|---------|------------------|-----------------|-------|
| *Jupi* | $3.08 \times 10^{-6}$ | 24.60 | 0.9991 |
| *Pérola* | $5.55 \times 10^{-6}$ | 26.25 | 0.9992 |

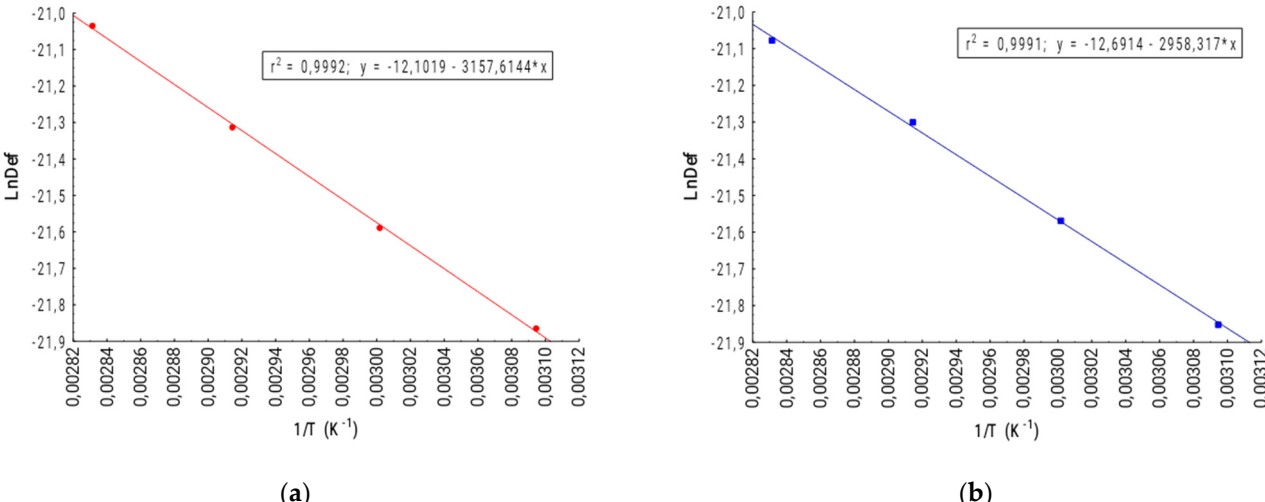

**Figure 2.** Representation of the effective diffusivity as a function of the inverse of the drying temperature of pineapple skins adjusted via an Arrhenius-type equation: (**a**) *Pérola*; (**b**) *Jupi*.

The activation energies ($E_a$) found for drying the skins of pineapples cv. *Jupi* and *Pérola* are lower than that reported by [59] in infrared drying of pineapple slices at temperatures ranging from 50 to 100 °C, with an $E_a$ value of 33.63 kJ/mol. The $E_a$ varies from product to product and is influenced by hygroscopicity, morphology, and environmental conditions [60]. Pineapple peels are within the activation energy range of agricultural products, from 12.7 to 110 kJ/mol, according to Zogzas, Maroulis, and Marinos-Kouris [61].

Table 7 shows the main thermodynamic properties obtained from pineapple peel drying. In both pineapple varieties, increasing the temperature leads to a decrease in enthalpy ($\Delta H$), indicating that higher drying temperatures require less thermal energy to remove water bound to the product during the drying process [62]. The values remained in the range of 21.9086 to 21.6592 kJ/mol for *Jupi* pineapple and 23.5656 to 23.3162 kJ/mol for *Pérola* pineapple. Close values were found by Almeida et al. [63] in the drying of achachairu (*Garcinia humilis*) peels at temperatures ranging from 40 to 80 °C with an air velocity of 1.5 m/s, with $\Delta H$ ranging from 24.96 to 24.63 kJ/mol.

**Table 7.** Thermodynamic properties of pineapple skins subjected to drying at temperatures from 50 to 80 °C.

| Variety | T (°C) | $\Delta H$ (kJ/mol) | $\Delta S$ (kJ/mol K) | $\Delta G$ (kJ/mol) |
|---------|--------|---------------------|----------------------|---------------------|
| *Jupi* | 50 | 21.9086 | −0.3511 | 135.3628 |
| | 60 | 21.8255 | −0.3513 | 138.8749 |
| | 70 | 21.7424 | −0.3516 | 142.3896 |
| | 80 | 21.6592 | −0.3518 | 145.9066 |
| *Pérola* | 50 | 23.5656 | −0.3462 | 135.4377 |
| | 60 | 23.4825 | −0.3464 | 138.9009 |
| | 70 | 23.3993 | −0.3467 | 142.3666 |
| | 80 | 23.3162 | −0.3469 | 145.8347 |

$\Delta H$—enthalpy; $\Delta S$—entropy; $\Delta G$—Gibbs free energy.

It is noticed that entropy ($\Delta S$), which measures the degree of disorder of the system, decreased with the increasing temperature, indicating less agitation of water molecules and a greater degree of order between these molecules [64]. Negative entropy values indicate the presence of chemical adsorption and/or structural modifications of the adsorbent, as well as that the drying processes are entropically unfavorable [65]. Similar behavior was also observed in the drying of achachairu (*Garcinia humilis*) peels [63], mesocarp of *baru* (*Dipteryx alata* Vogel) [66], and mesocarp of *pequi* (*Caryocar brasiliense* Cambess) [67].

The Gibbs free energy (ΔG) is the total amount of energy involved in the system, and positive values (ΔG > 0) mean that the drying process was not spontaneous; that is, it required external energy (heated air) for its effectiveness, resulting in the change in the liquid phase from water to steam [68,69]. In addition to the gradual increases brought about by the increase in drying temperature, the samples of the two pineapple varieties showed similar ΔG values, ranging from 135 to 145 kJ/mol, indicating that for the drying of both varieties, the total energy required is same. The increase in ΔG with the increasing temperature was also reported by Resende et al. [62], in the drying of *baru* fruits (*Dipteryx alata* Vogel) at temperatures from 40 to 100 °C, and by Morais et al. [70], on the drying of bacaba pulp (*Oenocarpus bacaba* Mart) at temperatures of 40, 50, and 60 °C.

Values close to ΔG were reported by Tavone et al. [71], varying between 127.4250 and 134.8996 kJ/mol for calabura fruits (*Muntingia calabura* L.) dried at temperatures from 40 to 60 °C.

The results of the centesimal composition of the two pineapple varieties are presented in Table 8, corresponding to the averages and standard deviations obtained for the in natura sample and the dehydrated samples at temperatures from 50 to 80 °C.

**Table 8.** Proximate composition of fresh peels of *Jupi* and *Pérola* pineapples and flours obtained from the peels after drying at temperatures ranging from 50 to 80 °C.

| Parameter | Variety | Drying Temperature (°C) | | | | |
|---|---|---|---|---|---|---|
| | | In Natura | 50 | 60 | 70 | 80 |
| Water content (% wb) | *Jupi* | 72.49 ± 0.32 bA | 9.57 ± 0.32 aB | 7.79 ± 0.27 aC | 7.80 ± 0.19 aC | 7.57 ± 0.46 aC |
| | *Pérola* | 73.42 ± 0.21 aA | 8.67 ± 0.33 bB | 7.36 ± 0.16 aC | 7.33 ± 0.10 bC | 6.36 ± 0.09 bD |
| Ashes (% db) | *Jupi* | 3.15 ± 0.09 bB | 3.52 ± 0.02 aA | 3.19 ± 0.08 bB | 3.30 ± 0.01 bB | 3.60 ± 0.15 aA |
| | *Pérola* | 3.65 ± 0.07 aA | 3.56 ± 0.03 aA | 3.48 ± 0.07 aA | 3.47 ± 0.13 aA | 3.64 ± 0.13 aA |
| Proteins (% db) | *Jupi* | 4.63 ± 0.01 bC | 4.77 ± 0.01 bA | 4.68 ± 0.01 bB | 4.68 ± 0.00 bB | 4.67 ± 0.00 bB |
| | *Pérola* | 4.78 ± 0.03 aD | 5.20 ± 0.01 aA | 5.12 ± 0.01 aB | 5.12 ± 0.01 aB | 5.08 ± 0.01 aC |
| Lipids (% db) | *Jupi* | 1.17 ± 0.02 bD | 2.22 ± 0.02 bBC | 2.26 ± 0.03 bB | 2.15 ± 0.03 bC | 2.35 ± 0.03 bA |
| | *Pérola* | 1.67 ± 0.04 aC | 2.42 ± 0.03 aB | 2.87 ± 0.01 aA | 2.87 ± 0.06 aA | 2.88 ± 0.02 aA |
| Carbohydrates (% db) | *Jupi* | 91.05 ± 0.13 aA | 89.49 ± 0.02 aC | 89.86 ± 0.07 aB | 89.87 ± 0.01 aB | 89.39 ± 0.12 aC |
| | *Pérola* | 89.89 ± 0.08 bA | 88.83 ± 0.03 bB | 88.53 ± 0.09 bC | 88.54 ± 0.08 bC | 88.40 ± 0.14 bC |
| TEV (kcal/ 100 g db) | *Jupi* | 393.21 ± 0.65 aC | 397.02 ± 0.10 aB | 398.53 ± 0.50 bA | 397.54 ± 0.29 bAB | 397.36 ± 0.28 bAB |
| | *Pérola* | 393.76 ± 0.75 aC | 397.87 ± 0.43 aB | 400.46 ± 0.19 aA | 400.44 ± 0.83 aA | 399.83 ± 0.57 aA |

db—dry basis; wb—wet base; TEV—Total Energy Value. Means followed by the same lowercase letter in columns and uppercase in rows do not differ statistically by Tukey's test at 5% probability.

The water content of the *Jupi* variety in natura was slightly higher than that of the *Pérola* variety, but with drying, the relationship was reversed, with *Jupi* contents resulting in lower values at most drying temperatures. Also, the highest water content of the flours occurred at a drying temperature of 50 °C and the lowest at 80 °C. The final water contents are considered adequate for the storage of agricultural products in general and, in particular, flours, whose values are considered satisfactory when lower than the maximum limit of 15% [72]. Similar water content values were studied in products such as kiwi peel flour [73] and tropical fruit residues [74].

The ash content in the in natura samples was higher in the *Pérola* variety, however, in a percentage difference that was not very expressive, so that in the flours, this difference only remained statistically significant in the samples obtained after drying at 60 and 70 °C. According to Storck et al. [75] the mineral content (ash) can be influenced by the grain size of the sample, which would explain the small variation in the values of the present research. The ash content detected in the flours was lower than those reported by Erkel et al. [76] and Santos et al. [77] on dehydrated pineapple peels, whose values were 4.70 and 4.66% db, respectively, and by Brito et al. [78], who reported a value of 6.45% db in pineapple crown flour.

The *Pérola* variety had the highest protein content in the in natura sample and this difference, in relation to *Pérola*, was maintained in all flours. The samples with the highest levels were obtained after drying at 50 °C, with values gradually decreasing with the increasing temperature. Although fruits do not represent a good source of protein, pineapple peel flours proved to be an alternative for incorporation into other products for protein purposes and may be a possibility of food for needy populations, which have difficult access to rich foods in proteins [75]. The results presented here are higher than the protein content of apple flour (2.7 to 3.5% db) [75] and close to the pineapple residues (peel and crown) of 4.56% before the fermentation process [79].

The flours produced showed low levels of lipids, with a slight advantage of the *Pérola* variety, in the in natura sample maintained in the flours obtained at all temperatures. Damasceno et al. [80] obtained lower levels (1.17 ± 0.08%) in *Pérola* pineapple peel flour, as well as Vieira et al. [81] on the dehydrated residues of pineapple (0.78% db) and cashew (1.63% db). Low levels of lipids in flours contribute to the enhancement of the product, given that lipids have a high caloric content, and their concentration is taken into account in products with a functional purpose or reduced caloric value [82].

The highest proportion in the composition of the samples, as expected, is composed of carbohydrates, with higher values in the in natura sample of the *Jupi* variety, but with a relatively small difference for the *Pérola* sample. The effect of temperature was not very expressive and did not result in consistent variations. The high values of total carbohydrates include dietary fiber, which is a percentage of important components of fruit peels, which, according to Garcia-Amezquita et al. [83], the total dietary fiber contents of orange, mango, and prickly pear peels dried via convection and freeze-drying correspond to 54.7, 54.8, and 49.2% db, respectively. Carbohydrate values lower than pineapple peel flours were quantified by Reis et al. [84], in passion fruit and orange peel flours with contents ranging between 72.12 and 73.37% db, and by Silva et al. [85], in avocado seed flour with a value of 79.06% db. Fresh pineapple peels had higher carbohydrates than tropical fruit pulp residues (guava, acerola, pineapple, soursop, *bacuri*, and *cupuaçu*) with values of 81.20, 65.02, 80.78, 77.14, 64, 41, and 9.77% db, respectively [86].

Observing the results of the Total Energy Value (TEV), the flours had a higher value when compared to the in natura samples, explained by the lower water content of the samples and, therefore, the higher energy density. In addition, flours had higher levels of proteins and lipids, also contributing to the increase in calories. The calorific results obtained for the flours were higher than those found for wheat flour (360 kcal/100 g) and slightly lower for soy flour (404 kcal/100 g) [87]. *Pequi* peel flour (225.42 kcal/100 g db) was also superior [88].

The results of the physicochemical characterization of the two pineapple varieties are presented in Table 9, corresponding to the averages and standard deviations obtained for the in natura sample and the dehydrated samples at temperatures from 50 to 80 °C.

**Table 9.** Physicochemical characterization of the in natura peels of *Jupi* and *Pérola* pineapples and the flours of the peels obtained after drying at temperatures from 50 to 80 °C.

| Parameter | Variety | Drying Temperature (°C) | | | | |
|---|---|---|---|---|---|---|
| | | In Natura | 50 | 60 | 70 | 80 |
| Water activity (25 °C) | *Jupi* | 0.984 ± 0.001 aA | 0.291 ± 0.00 bB | 0.278 ± 0.00 aC | 0.241 ± 0.00 bD | 0.235 ± 0.00 bD |
| | *Pérola* | 0.988 ± 0.001 aA | 0.305 ± 0.00 aB | 0.270 ± 0.00 bC | 0.252 ± 0.00 aD | 0.243 ± 0.001 aD |
| Acidity (% citric ac. db) | *Jupi* | 2.66 ± 0.02 aBC | 2.59 ± 0.10 bC | 2.71 ± 0.10 bBC | 2.83 ± 0.10 bAB | 2.99 ± 0.10 bA |
| | *Pérola* | 2.13 ± 0.05 bC | 3.09 ± 0.10 aB | 3.10 ± 0.00 aB | 3.27 ± 0.00 aB | 3.64 ± 0.10 aA |
| Total sugars (g/100 g db) | *Jupi* | 45.10 ± 0.72 aE | 52.12 ± 1.27 aD | 54.92 ± 0.89 aC | 62.78 ± 0.16 aB | 71.73 ± 0.81 aA |
| | *Pérola* | 35.49 ± 0.31 bD | 41.38 ± 0.09 bC | 41.95 ± 0.16 bC | 44.45 ± 0.09 bB | 48.81 ± 0.33 bA |
| Reducing sugars (g/100 g db) | *Jupi* | 41.45 ± 0.65 aA | 40.99 ± 0.09 aA | 40.81 ± 0.08 aA | 36.72 ± 0.21 aB | 32.33 ± 0.97 aC |
| | *Pérola* | 32.78 ± 0.46 bC | 36.84 ± 0.56 bA | 35.68 ± 1.04 bAB | 35.05 ± 0.27 bB | 30.50 ± 0.12 bD |

db—dry base. Means followed by the same lowercase letter in columns and uppercase in rows do not differ statistically by Tukey's test at 5% probability.

The water activity ($a_w$) did not follow the water content in most samples, presenting statistically similar values between the in natura samples and statistically different values between the flours of the two varieties. A tendency in the reduction in aw with the increase in drying temperature was also observed. The low values of $a_w$ and the water content in the dehydrated samples are adequate to avoid biochemical and microbial reactions responsible for the loss of in natura products, making them suitable for storage [89–91]. Other studies addressing the drying of residues from fruit processing also correlated the reduction in $a_w$ with the drying temperature: peels and stalks of pineapple cv. *Pérola* dried at 50, 60, and 70 °C [92]; white-fleshed pitaya peels dehydrated at 50, 60, and 70 °C [93]; and acerola agro-industrial residues dried at 60 and 80 °C [94].

The acidity content was higher in the *Pérola* variety in the flours obtained at all temperatures, surpassing the *Jupi* variety samples by about 0.5% or more. The increase in drying temperature resulted in a trend towards increased acidity in all samples. Similar behavior was also verified in taro flour with acidity contents of 0.77, 0.92, and 0.98% db for drying temperatures of 70, 80, and 90 °C, respectively [95].

The total sugar content was higher in the *Jupi* variety, thus remaining in the samples obtained at all drying temperatures. With the increase in temperature, the contents also increased, with increases between 37 and 57% between in natura and dried samples at 80 °C in the *Jupi* and *Pérola* varieties, respectively. The increase in the sugar content of the material subjected to drying can be attributed to a possible hydrolytic activity caused by the effect of time and temperature [96]. A similar behavior was described by Alves, Machado, and Queiroga [97] when they obtained higher levels of total sugars in cashew apple flour when compared to the in natura sample. The total sugars of the pineapple peel flours were much higher than that quantified by Queiroz et al. [98] in lychee peel flour with an average value of 11.08 g/100 g db.

The reducing sugars initially accompanied, like the total sugars, the highest value for the *Jupi* variety, surpassing the value of the *Pérola* in natura sample by more than 25%. With the drying, the behaviors diverge, reducing the values consistently in the *Jupi* variety between 50 and 80 °C, while in *Pérola*, an increase was observed among the in natura sample and the others, except for the sample obtained at 80 °C. According to Santos et al. [93], the reduction in reducing sugars with heating can be explained by the Maillard reaction, which results in the degradation of reducing sugars when complexing with free amino acids.

## 4. Conclusions

Of the ten drying kinetics adjustment models used for pineapple residues, nine showed good prediction, with emphasis on the Diffusion Approximation, Page, and Midilli models.

The effective diffusivity values were between $3.1 \times 10^{-10}$ and $7.3 \times 10^{-10}$ m$^2$/s, with similar results for the samples of both varieties and a good correlation with temperature via the Arrhenius-type equation, with a higher value of activation energy in the sample of the *Pérola* variety; the residues of this variety showed greater enthalpy and entropy variation in the drying process.

The residual pineapple peels of the *Jupi* and *Pérola* varieties were sources of minerals, sugars, and proteins, with higher levels of acidity and proteins in the *Pérola* variety and more sugars in the *Jupi*.

**Author Contributions:** Conceptualization, C.G.d.R., R.M.F.d.F. and A.J.d.M.Q.; data curation, Y.F.P., F.S.d.S., T.L.B.d.L. and L.T.S.A.; formal analysis, C.G.d.R., Y.F.P. and D.d.C.S.; investigation, C.G.d.R., L.T.S.A. and F.S.A.; methodology, C.G.d.R. and R.M.F.d.F.; software, J.P.d.L.F., J.P.G. and W.P.d.S.; supervision, R.M.F.d.F. and A.J.d.M.Q.; writing—original draft, C.G.d.R.; writing—review and editing, R.M.F.d.F. All authors have read and agreed to the published version of the manuscript.

**Funding:** This research received funding from FAPESQ—Foundation and Support for Research of the State of Paraíba (Brazil).

**Institutional Review Board Statement:** Not applicable.

**Data Availability Statement:** Data can be digitized from the graphs or requested from the corresponding author.

**Acknowledgments:** The authors would like to thank the Federal University of Campina Grande (Brazil) for the research infrastructure and FAPESQ—Foundation and Support for Research of the State of Paraíba (Brazil) for funding the research. The first author would like to thank the Coordination for the Improvement of Higher Education Personnel-Brazil (CAPES) for supporting this study and for the research grant.

**Conflicts of Interest:** The authors declare no conflict of interest.

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
