# Peer review of "Pineapple Peel Flours: Drying Kinetics, Thermodynamic Properties, and Physicochemical Characterization"

_processes, doi:10.3390/pr11113161_

Round 1
Reviewer 1 Report
Comments and Suggestions for Authors
Research on Pineapple peel flours: drying kinetics, thermodynamic properties and physicochemical characterization, is fascinating. However, there are several suggestions to improve the quality of this manuscript:
1. In lines 16–17, the problems expressed in sentences 1 and 2 are less correlated with the desired research objectives
2. In the introduction, paragraphs 1 and 2 need more information, and the sentences must be longer.
3. The urgency and novelty of the research have yet to be stated clearly in the introduction. Complete it with research that has been carried out and its weaknesses, then update it in this research.
4. In the introductory part, the problem to be researched has yet to be explicitly stated.
5. On line 81, complete the information regarding the characteristics of a ripe pineapple.
6. The method for determining physicochemical characteristics should be described in the method and supplemented with references.
7. The discussion should be more in-depth and comprehensive in discussing the phenomenon of each data obtained
Comments on the Quality of English LanguageExtensive editing of English language required
Author Response
Dear reviewer
We appreciate your contributions to improving the quality of our manuscript.
We would like to inform you that all revisions have been carried out.
We are available for questions or suggestions
Yours sincerely
Reviewer 2 Report
Comments and Suggestions for Authors
This study aimed to prepare flours from the peel of pineapples of the Pérola and Jupi varieties from convective drying at temperatures of 50, 60, 70 and 80 °C, to determine the drying kinetics, to adjust mathematical models to the experimental data, to determine the effective diffusivity and thermodynamic properties and to characterize the flours obtained. However, there are still the following problems:
1. Carefully check the grammatical errors in the article and polish the language.
2. Please explain what are the current pineapple peel flours drying methods? Combined with the existing problems, the purpose of this study is analyzed.
3. Explain why convection drying is used to dehydrate pineapple peel flours ?
4. Please provide a schematic diagram of the test equipment
Comments on the Quality of English Language1. Carefully check the grammatical errors in the article and polish the language.
Author Response

(The authors gave the same response as above.)

Reviewer 3 Report
Comments and Suggestions for Authors
Dear Authors and Editor,
The research is interesting and fits into contemporary research trends because it describes the process of producing flour from waste, i.e. pineapple peels. The authors should explain in more detail what new contributions the work brings to the existing literature
Introduction
In introduction section authors should mention that such flour has already been tested in innovative food products, e.g. cereal bars, muffins, biscuits - The authors should mention it in their research and cite the relevant literature. The effect of drying on the physicochemical and techno-functional properties of pineapple (Ananas comosus) peel flour was described by Lopez-Nunez et al. (2017) - please explain what new this current research brings.
Lopez-Nunez, J. S., Salcedo-Mendoza, J. G., … Perez-Sierra, O. A. (2017). Effect of Drying on the Physicochemical and Techno- Functional Properties of Pineapple Peel Flour. Indian Journal of Science and Technology, 11(46), 1–7. https://doi.org/10.17485/ijst/2018/v11i46/137450
Materials and methods
In material and method section please provide some technical parameters of the manufacturer's pulper, city and country. Please provide the approximate storage time for pineapple peel pulp in the freezer. Please explain whether the pineapple peels were thawed before drying and what was the defrosting process like? Regarding an oven used for drying please add some technical parameters and type, manufacturer, kind of drying should be mentioned.
Table 1
Please check if the expression in the equations exp exp is correct, I’m not sure but it seems that there should be only one exp? RX – do you mean MR? “moisture ratio” as usually used in literature.
Author Response

(The authors gave the same response as above.)

Round 2
Reviewer 2 Report
Comments and Suggestions for Authors
no
Comments on the Quality of English LanguageMinor editing of English language required
Author Response
Dear reviewer
The language has been revised to the extent possible.
Thank you again for your comments to improve our manuscript.
Best regards
Reviewer 3 Report
Comments and Suggestions for Authors
Dear Author,
Please confirm if "exp exp" entered twice in the formulas are correct because in the literature there is only one exp in the same formulas. I'm not sure and you didn't answer this question for me earlier.
Best regards,
Author Response
Dear reviewer
I apologize for the mistake, but now the equations have been revised.
Thank you once again for your comments to improve our manuscript.
Best regards